# Applying Spatial Video Geonarratives and Physiological Measurements to Explore Perceived Safety in Baton Rouge, Louisiana

**DOI:** 10.3390/ijerph18031284

**Published:** 2021-01-31

**Authors:** Alina Ristea, Michael Leitner, Bernd Resch, Judith Stratmann

**Affiliations:** 1Boston Area Research Initiative, School of Public Policy and Urban Affairs, Northeastern University, Boston, MA 02115, USA; a.ristea@northeastern.edu; 2Department of Geography and Anthropology, Louisiana State University, Baton Rouge, LA 70803, USA; mleitne@lsu.edu; 3Department of Geoinformatics, University of Salzburg, 5020 Salzburg, Austria; 4Center for Geographic Analysis, Harvard University, Cambridge, MA 02138, USA; 5Spatial Information Management, Carinthia University of Applied Sciences, 9524 Villach, Austria; judithstratmann1@gmail.com

**Keywords:** perceived safety, spatial video, geonarrative, moments of stress, mixed-method approach, wearable sensors, spatiotemporal semantic analysis, sentiment analysis

## Abstract

Spatial crime analysis, together with perceived (crime) safety analysis have tremendously benefitted from Geographic Information Science (GISc) and the application of geospatial technology. This research study discusses a novel methodological approach to document the use of emerging geospatial technologies to explore perceived urban safety from the lenses of fear of crime or crime perception in the city of Baton Rouge, USA. The mixed techniques include a survey, spatial video geonarrative (SVG) in the field with study participants, and the extraction of moments of stress (MOS) from biosensing wristbands. This study enrolled 46 participants who completed geonarratives and MOS detection. A subset of 10 of these geonarratives are presented here. Each participant was driven in a car equipped with audio recording and spatial video along a predefined route while wearing the Empatica E4 wristbands to measure three physiological variables, all of them linked by timestamp. The results show differences in the participants’ sentiments (positive or negative) and MOS in the field based on gender. These mixed-methods are encouraging for finding relationships between actual crime occurrences and the community perceived fear of crime in urban areas.

## 1. Introduction

Research has consistently indicated that actual crime occurrences and the perceived fear of crime in urban areas are key concerns for society and that safety is highly important for a sustainable environment [1]. At the same time, it has been discovered that fear of crime tends to occur at higher rates than reported crime occurrences [2,3]. One study showed that the perception of safety in a campus environment correlated with objective crime reporting, which is debatable at a larger city scale [4]. An element that can be influential for crime perception is urban blight or urban decay. Urban blight refers to the phenomenon when a part of a city becomes deteriorated due to neglect for various reasons, mainly related to socio-economic conditions. Signs of blight are abandoned buildings or desolate areas, which are highly visible only at the micro-scale.

The application of geospatial technologies in researching spatial crime patterns started in the USA in the second half of the 1980s [5]. During the same period, the safety of urban neighborhoods became the focus of many studies. Jacobs and Newman were pioneers in the study of urban planning concerning issues of safety [6,7], contributing to the creation of a new sub-discipline in criminology called Crime Prevention Through Environmental Design (CPTED) [8]. CPTED is based on an agenda considering the proper design of the built environment and the effective use of buildings and public spaces in neighborhoods for diminishing fear and incidence of crime. Overall, CPTED methodologies lead to an improvement in the quality of life for citizens. Many researchers are contributing to the development of CPTED from various fields such as psychology, geography, and criminology. For example, Ref. [9] encouraged environmental criminology with the theory of crime patterns, including crime attractors, generators, and detractors, which influenced discussions of various attractors for safety–crime perception. The Broken Windows Theory [10] is another contribution stating that visible signs of public disorder, and anti-social behavior, create an urban environment more prone to crime. Since the 1970s researchers and practitioners started incorporating the social aspect in the CPTED, which was considered limited to the built environment.

One interdisciplinary research path includes the use of individual data nested in neighborhoods or similar clusters through hierarchical modeling, to examine connections between the built environment, social elements, and perceived crime [11,12,13,14]. These analyses are generally done by using social observation surveys. Ref. [15] used a multilevel analysis in the city of Chicago to analyze collective efficacy, defined as social cohesion among neighbors, combined with their willingness to intervene on behalf of the common good, and applies the social observation method to observe disorder in urban neighborhoods [11]. The authors [16] found that in neighborhoods where the cohesion among residents and the social observation of police officers is high, the rate of violence is low. Additionally, they remark that physical and social disorder decrease when collective efficacy is strong. Ref. [13] discusses the social environment differences, finding that White residents consistently perceive more crime or disorder than their neighbors. This study [14] conducted mediation analyses to examine the effects of gender and the presence of park use for different activities, based on the perception of crime (i.e., women use parks less for physical activity than men, and explanations include safety reasons). Complimenting this, our fusion approach supports the data collection in the field together with using the social observation surveys in the office or classroom, for a better social and environmental experience.

Researchers discussing the CPTED and their interventions are interconnected with elements of fear of crime, mostly because CPTED is about reducing crime and fear of crime. Studies show that fear of crime arises from community disorder, and it is based on the social and physical environment [17]. Moreover, multiple factors influence fear such as the sense of vulnerability, gender, physical and social blight, group conflicts, cultural background, and others [18]. Additionally, research shows that, when splitting the categories of residential land use, the condominium and the apartment areas show the highest levels of perceived unsafety [19,20]. Hence, the quality of life is affected by crime, fear of crime, and anti-social behavior, which are valuable indicators in people’s choices of a place to live [21]. Considering the large volumes and variety of dynamic data, and the changes in human behavior, there is a need for more detailed and complex information regarding urban safety and security, whether subjectively or objectively collected.

The general aim of this research is to document the use of emerging geospatial technologies to explore perceived urban safety from the lenses of fear of crime or crime perception. This work is a proof of concept on combining quantitative and qualitative methods and techniques. Such technologies are also able to collect contextual information in a standardized way and in a format that can be archived so that these technologies can be used in the long term and also for future comparative studies. In this research, the contextual information refers to a narrative, defined as an interview that is conducted with a test subject, collected with an audio recorder, and later transcribed into text.

This research combines different geospatial technologies, including systematic video data acquisitions, geographical storytelling, and human physiological measurements, which together allow the analysis of an urban environment through a GIS-based platform. The two main objectives of this research are to test the compatibility of data acquisition through mixed geospatial technologies, and to extract safety information from the data acquired using mixed methods and to implement it in a GIS-based model. Another component of this work refers to discussing the comparison between official crime data reported to the police, and peoples’ perceived safety that were all collected, extracted, and analyzed from the mixed-method approach.

## 2. Data and Methods

This research is based on the handling of data acquired from different sources, including online and self-collected through a questionnaire/mapping exercise, and a field survey leveraging a suite of geospatial technologies.

### 2.1. Data

#### 2.1.1. Study Area

The study area selected for this research is the city of Baton Rouge, the state capital of Louisiana, USA (Figure 1). Together with three other smaller cities (Baker, Central, and Zachary), Baton Rouge is part of the East Baton Rouge Parish (EBRP). Baton Rouge consists of 58 neighborhoods and occupies an area of 123.84 km^2^ on the east bank of the Mississippi River. According to the most recent census data from 2010, Baton Rouge has a population of 229,493. The racial composition includes 54.8% African Americans and 36.6% Whites.

#### 2.1.2. Online Data Sources

Reported crimes from 2018 collected by the Baton Rouge Police Department (BRPD) that included the crime location and the time of the crime occurrence were retrieved online. Initially, the total number of crimes located in the EBRP was 45,561. After geocoding and cleaning, this number was reduced to 44,964 crime records. Of these, 44,554 crimes were located inside the city limits of Baton Rouge. The crime types include assault, battery, burglary, criminal damage to properties, firearm, homicide, narcotics, nuisance, robberies, and theft. Additional data collected for this study included 311 calls for service and a series of data from the 2010 US Census (spatial resolution: census block groups) namely residential population, ethnicity, education, household types, foreign-born, unemployment, poverty rate. Other data (street network, building footprints, public buildings, and neighborhoods) was collected from the Open Data Baton Rouge portal.

#### 2.1.3. Questionnaire with a Mapping Exercise

The purpose of the questionnaire and the on-screen mapping exercise, implemented in Google Maps, was to collect baseline information about the perceived safety in Baton Rouge and the role that urban blight plays in this perception. Both the questionnaire and the mapping exercise were set up online, and participants were expected to complete the forms and submit their responses to the authors of this research. The mapping exercise required test participants to draw polygons, lines, and/or points into a map of the city of Baton Rouge where they would feel unsafe, and to provide a short description of the perceived feelings. Results from the analysis from the mapping exercise were not included in this research. The questionnaire and the mapping exercise each lasted for about 10 min, and details can be found in [22].

#### 2.1.4. Field Survey

The field survey involved driving with a vehicle along a selected route with test participants. Additional occupants of the car included a driver and one or two more researchers acting as navigator and interviewer. Occupants were also tasked with observing the geospatial technology equipment to make sure that it was working properly [23]. The test participant was always sitting in the front next to the driver. The selection of the route was crucial for this research, and selected based on the above-mentioned questionnaire, a prior survey, and other criteria, which are discussed next. First, regarding the results of the questionnaire, the field survey should follow a route that is partly located in areas where people’s safety perception was indicated to be low and high. Second, in a prior survey project, physical urban blight indicators were collected in a total of five Baton Rouge neighborhoods by applying the spatial video technology. That survey resulted in the collection of 1717 urban blight locations, with each location showing one or more blight indicators [22]. Subsequent correlation analysis indicated that urban blight indicators showed a medium to a high statistically significant positive correlation with different crime types. This means that areas with high incidents of urban blight would possess high crime rates, and vice versa [22]. Based on this prior survey and correlation results, the selected route in this research should be located in areas of high, medium, and low physical urban blight incidents and crime rates, respectively. Third, additional criteria for the route selection were that (a) the driving time should last for around 30 min, (b) located in residential areas with (c) diverse economic status, and (d) not including highways and lakes. The final route selected is shown in Figure 1, and it is located almost in the middle of the city of Baton Rouge.

#### 2.1.5. Geospatial Technology

The data collected during the field survey proceeded with three different types of geospatial technology instruments, including spatial videos, (audio) recording devices, and wristbands measuring galvanic skin response (GSR) and skin temperature (ST). Each of these technologies is briefly introduced next.

##### Spatial Video Technology

Streaming videos were collected using the Spatial Video Acquisition System (SVAS), or spatial video, which allows gaining an on-site point of view and can be applied to various types of research that require impartial visual information [24]. A Global Positioning System (GPS) receiver is integrated into each digital video camcorder, which allows the collection of spatially referenced digital video material. Furthermore, a timestamp is attached to each video frame. Unlike Google Street View, SVAS data collection is in the control of the researcher. Spatial videos can be collected using a variety of modes (car, motorbike, bicycle, boat, and by foot). This technology can record videos from a survey vehicle in the direction of travel with one or more cameras [25]. However, having cameras just in the front can be insufficient, because they cannot detect the side angles. This is the reason why additional cameras may be recommended. Currently, no standards exist regarding video format, resolution, or type of storage database. In prior research, spatial video technology was successfully applied to various fields.

For example, in [23,24], the SVG is used to illustrate how individuals are coping both during the disaster and recovery phase and support spatially targeted interventions. By using this technology, there are multiple perspectives for the same geographic area and this can help health officials or policymakers to see how the residents see the situation in the field. Another interesting example from [23] includes the connection, or lack thereof, of drugs perceptions between police and community members. Overlaying professional insights on the drug problems (from law enforcement) and local insights (from the community and nonprofits) can help place-based social interventions.

In [26], researchers used spatial video technology as a data acquisition method for roadside advertisements, revealing patterns of their spatial distribution. The highest advertising density in the study area was associated with bends and intersections, mostly including advertising flags. Less than 2% of these advertisements changed their content, which can be a higher potential to distract drivers than normal boards. In a different study, researchers used spatial video for collecting soccer-related graffiti and metadata from the images [27]. The results show hotspots of positive graffiti for the three rival teams in Krakow, Poland, hotspots of hate for each team, and areas of conflict (i.e., teams graffitied one design over another). Another application includes pattern analysis of secondhand smoke exposure on college campuses [28]. Spatial videos are collected during times of heavy pedestrian traffic on bicycles. Finding hotspot locations of observed smoking is useful for college administration, policymakers, and health officials—with these types of data they can better understand outdoor smoking patterns on campus and, for example, create disperse locations where smoking is allowed.

In this research study, spatial video recording was conducted with the use of five Contour+2 Action Cameras (Model 1700), four of which were attached to the backseat windows (two on each side) of the vehicle. These cameras provided a frontal view of the left- and right-hand side of the street’s adjacent properties, perpendicular to the driving direction. The fifth camera was mounted on the front windshield, pointing towards the driving direction. All cameras were attached to the inside windows of the vehicle, making the cameras almost invisible from the outside (Figure 2). Cameras included a 170-degree wide-angle lens, favorable for recording narrow streets. Cameras also contained an internal microphone, however, an external microphone or an audio recorder was used. This eliminated noise effects inside the vehicle environment (see the section on “Geonarrative approach” below). Videos were recorded in high definition (1080 p) and collected 30 frames per second. Videos were saved on memory cards that could store a high-definition video file with a maximum size of ~4 GB. This translated to approximately 40 min of driving time.

##### Geonarrative Approach

As stated before, a narrative refers to an interview that is conducted with a test subject, collected with an audio recorder (Figure 3), and later transcribed into text. Each word, phrase, or sentence of the interview can be associated with a timestamp during the audio recording. When the interview takes place inside a vehicle during spatial video data collection, the timestamp associated with each word, phrase, or sentence of the interview can be matched to the timestamp of each video frame and attributed to the GPS location of that frame. That allows each word, phrase, or sentence of the interview (after being transcribed in a certain format) to be put on a map at the exact location, where it was mentioned during the interview. The mapped interview is referred to as a geonarrative and the integration of spatial video and geonarrative data collection is called a Spatial Video Geonarrative (SVG).

SVG was applied in this study since safety perception and urban blight may be strongly associated with specific locations and their environments inside urban neighborhoods. While the same interview with test subjects could have been conducted in an office space, interviewing as part of an SVG has one major benefit, namely, stimulating the discussion with test subjects on topics related to the environment (e.g., safety perception) that they are driving through [9]. This may also stimulate test subjects’ recollection and memory about the interviews’ main topics. The role of the environment is thus to enrich a geonarrative with contextual information, that is likely impossible when the interview is conducted in an office space. Test subjects’ opinions and experiences may also be easier triggered, when conducting a geonarrative, further enriching the interviews’ content. In this study, geonarratives were based on mostly unstructured interviews due to test participants’ having a wide range of expertise and familiarity with the surveyed area. To collect as much information as possible, the interview was semi-structured, namely, there were a few questions repeated for each subject, and the rest of the interview was conversational, letting the subject express their feelings and beliefs. This is a qualitative approach, through which we could obtain more in-depth information about participants′ experiences. An SVG was completed with all test subjects while traveling along the same route as depicted in Figure 1. The use of an SVG permits finer spatial-scale data collection, capturing more locations of relevance than other methods (for example, paper surveys taken in the classroom). SVGs also provide the plus of spatial-temporal processes and possible connections linked to specific places. The flexibility in space and time included in the combination of spatial video and geonarratives, can be used for collecting data on the geographic context of many phenomena. The metadata from both, besides location and time, includes environment images, video captions, raw text (from the interviews), leading to sentiments and emotions analysis, topics, clusters of topics in space and time, longitudinal analysis (e.g., for studying gentrification).

##### Wearable Physiological Sensors

Empatica E4 wristbands were used in this research to measure physiological attributes from each test participant during the SVG (Figure 4). Physiological data collected from each participant included: galvanic skin response (GSR), which measures electrical properties of the skin; skin temperature (ST); blood volume pulse (BVP); and 3-axis acceleration, which captures motion-based activity. In addition, timestamps associated with physiological measurements were also collected and, similar to geonarratives, matched to corresponding timestamps from spatial videos and located in maps. In recent years, connections have been found between these measurements and stress detection and recognition, office environments [29], people with dementia [30] and physiological signal-based emotion recognition [31]. The physiological measurements of GSR and ST were input into a newly developed algorithm to detect test participants’ moments of stress (MOS) [32]. The algorithm greatly relies on relative measures to account for the individual conditions and characteristics of the test persons. The MOS algorithm is a rule-based algorithm based on galvanic skin response and skin temperature. It combined knowledge from empirical findings and expert information to ensure transferability between laboratory settings and real-world field studies [32]. As an input it uses the data from the Empatica E4 wristband (or other wristband or tool measuring the same features in the same parameters). The output is a .csv file or a similar type, with MOS score, timestamp, and location (if an additional tool for determining location is used together with the wristband).

### 2.2. Methodology

Standard graphical displays, such as bar and pie charts, were designed to explore test participants’ responses to the questionnaire. These results served as base information in this research study to understand how people feel, in general, about perceived safety issues in the city of Baton Rouge and the role that urban blight plays in this perception. As mentioned above, each spatial video file had embedded a GPS track. The software “storyteller” from Contour was used to extract the GPS track and to visualize it in a GIS. While during the past few years a growing body of the literature has started to include the creation and analysis of geonarratives, only recently have researchers began to build a designated software, such as Wordmapper [33], that was applied in this research. This software is user friendly and intuitive, combining information from GPS tracts and transcribed interviews matched by corresponding time stamps. Some basic contextual analysis of the geonarrative can also be accomplished in the current version of Wordmapper (Figure 5). Of each participants’ geonarrative, audio files were manually transcribed to the text and specifically formatted to be input into Wordmapper. This included, for example, to add a specific time format, i.e., (00:00:00) and to remove irrelevant words, e.g., “aaaa”, “pffff”.

The spatial distribution of reported crimes, and urban blight incidences were interpolated with kernel density estimations (KDEs) and visualized in a GIS. KDE is a non-parametric algorithm used to estimate the probability density function of a random variable. It is a fundamental interpolation method for spatially discrete data. KDE results depend on the set of three different parameters, including the kernel function type, each function’s bandwidth, and the cell size [34]. KDE is also used to interpolate the negative and positive polarities of the transcript text. For visualizing the MOS distribution, we applied a custom spatial distribution method developed by [35], which considers the MOS ratio for the geo-located detected MOS.

Sentiment analysis is one of the most important applications of natural language processing (NLP), the study of the extraction of opinions and feelings from the text. Generally, sentiment analysis tools rely on lists of words and/or phrases with positive and negative values. In the past few years, many dictionaries of positive and negative words have been developed. For example, Liu and Hu’s opinion lexicon contains around 6800 positive and negative opinion words or sentiment words for the English language [36]. AFINN, 2009–2011, is an example of a human-labeled lexicon, containing a list of English words rated for valence with an integer between minus five (negative) and plus five (positive) [20]. In this research, a Python package called Vader sentiment was applied. This is a lexicon and rule-based sentiment analysis tunned for sentiments at the conversation level. It is open-source and it considers word order and degree modifiers. Specifically, the compound score was used, which was computed by summing the valence scores of each word in the lexicon, adjusted according to the rules, and then normalized to be between −1 (most extreme negative) and +1 (most extreme positive). This metric shows a normalized composite score measure of sentiment per sentence. Thus, positive sentiment is considered with a compound score ≥ 0.05, a neutral sentiment with a compound score >−0.05 and <0.05, and negative sentiment with a compound score ≤ −0.05. Results presented in this work only show the positive and negative values. Additional results from text analysis are represented by word clouds and bar charts of the most frequent words. These types of visualizations are important to highlight differences between various test participants.

Sensor-based emotion recognition can contribute to a better understanding of participants’ emotions, especially stress emotions. Multiple studies exist that attempted to detect, whether a participant is stressed or not. Machine learning elements were applied in the majority of these studies. The algorithm used in this research was developed by [32]. It is a rule-based algorithm based on galvanic skin response and skin temperature, implemented in the R programming language. Rules, instead of a machine learning algorithm were used, due to the ability to integrate information from experts and of a better understanding of processes [32]. This algorithm shows a high accuracy during the validation process compared to other alternative algorithms, which was the main reason why it was used for the present research.

## 3. Results

In this section, preliminary results from the questionnaire are discussed first. This is followed by the text analysis outcomes of geonarratives (sentiment and topic modeling). Finally, moments of stress are extracted from skin conductive wristband measurements and compared with the part of the spatial video and the geonarrative taking place at the same time.

### 3.1. Results of the Questionnaire

The results of the following discussion are based on a total of 44 fully completed questionnaires. Test participants were graduate and undergraduate students in the Department of Geography and Anthropology and the Department of Sociology at Louisiana State University in Baton Rouge, USA. The vast majority of participants lived in Baton Rouge (42 of the 44) of which 39 participants were US citizens and three were non-US citizens. The gender distribution among participants was 26 females and 18 males. Regarding the question “Where do you feel less safe?”, many respondents mentioned gas stations and alleys (29 out of 44, each), and under bridges or underpasses (24 out of 44) (Figure 6). Additional places added to the provided list were the Louisiana State University (LSU) campus, parking garages, bars, Tigerland, downtown areas, Alvin Dark, Ubers, smoke stores, banks, shopping malls, movie theaters, nightclubs, bars, and places that serve alcohol.

Test participants’ crime perceptions were mostly influenced by crime hot spots with an average score of 3.83 (1 means “not at all influenced” and 5 means “most influenced”) and least influenced by social media information with an average score of 2.59 (Figure 7). While six out of the 44 respondents had previously been victims of crimes, including sexual assault, apartment, and car break-ins, and armed robbery, results from the questionnaire indicated that only one of these six participants felt very unsafe in Baton Rouge.

Around two-thirds of the respondents (66.1%) typically travel in Baton Rouge by car (Figure 8). This is not very surprising, due to a poorly developed public transportation system (only bus routes), an almost complete absence of bike paths, and a very small portion of roads having pedestrian walkways. Besides, Baton Rouge is a widely spread out city, making it difficult to travel without a car. A total of 83% of test participants’ safety perception was “moderately” to “highly” influenced by urban blight conditions (Figure 8). While Baton Rouge is considered a high crime city, 50% of respondents feel moderately safe in the city (Figure 8).

### 3.2. Results of Geonarratives

The majority (32 out of 44) of graduate and undergraduate students from the Department of Geography and Anthropology and the Department of Sociology, who completed the questionnaire and mapping exercise, also participated in the geonarrative. Nine additional participants in the geonarrative part of this research were local stakeholders and an additional five represented experts in urban blight and criminal activity issues. This resulted in a total number of 46 test subjects, who completed the geonarrative. Participants had different nationalities and ethnic backgrounds, however, the majority held US citizenship and their ethnic background was White. Among the comments from the participants, we extracted the following examples as related to crime or blight issues: “Looks like people left the neighborhood due to unsafe circumstances.”, “Here it looks dirty and abandoned”, “I would not like to be in this neighborhood at night time”, “It is easier for thieves to hide in trees waiting to assault you”, “Homeless people tend to ask for money late at night—I get very insecure”. The discussion of the results is based on a subset of 10 participants, who represent students, five of them males and five of them females. Each of the geonarrative participants traveled along the same route, starting at the same location, traveling in the same direction, at a similar time of the day, and under similar weather conditions. The selected route traverses through predominantly African American neighborhoods located in the middle and northern parts (Fairfields, part of Mid City) of the route and one predominantly white neighborhood in the south (i.e., Garden District). During recent decades, the boundary between the two racially diverse neighborhoods has shifted north–south between Government St., North Blvd., and Florida Blvd. The current boundary may be slightly north of Government St, but south of North Blvd. The current boundary is visible, when physical urban blight occurrences, collected in prior research [8], are overlaid on top of the route with lower blight densities found in the south and higher densities located in the center and the north (Figure 9, left). However, this boundary is obscured, when reported crime densities, together with the route, are shown (Figure 9, right).

Sentiment analysis models were applied to analyze such text transcripts of the ten participants’ geonarratives. As mentioned above, individual sentences from a transcript can be visualized on the map at the location, where the sentence was recorded (Figure 10). For example, the sentence “I would never walk here alone” is classified as negative, similar to the sentence “Yeah if I’m alone I feel kinda uneasy about it”. In contrast, “The houses are much bigger and there’s no trash in the front yards. Everything is really nice. They obviously take care of their house” is showing a clear positive polarity.

Feelings expressed in these sample sentences are common for most of the participants. When traveling north of Government St., participants would mention not well-maintained houses and gardens, abandoned houses, trash, and many other signs of urban blight. However, the south of Government St. (i.e., Garden District) was perceived by participants to be clean and have a nice environment. At the same time, all participants discussed the apparent income gap between the two areas along the route (north and south of Government St.), which are spatially contiguous to each other.

Sentiment analysis results from each of the ten geonarrative transcripts were aggregated and mapped with KDEs (Figure 11). Interestingly, positive sentences show higher densities compared to negative sentences. This may be due to the type of sentiment analysis algorithm that was selected, and an alternative algorithm would have shown somewhat different results. Unexpectedly, both positive and negative polarities seem to be concentrated along the route north of Government St. It is worth mentioning that the sentiment algorithm tends to find more positive polarity than negative. However, the Garden District only exhibited high densities of positive polarity, but completely lacked negative polarity. The Garden District is a residential neighborhood located in Baton Rouge′s Mid-City area and it is a well known historic area with various upscale homes and an active civic association. As considered by the geonarratives participants and Baton Rouge inhabitants, it is a safe neighborhood (confirmed by the low crime rates).

There are many mixed sentiments during the geonarratives for which we are planning on creating categories or applying topic modeling as future work. In the following rows are parts of geonarratives, which can be mapped in space and time (A is the respondent and Q is the interviewer):
A: I am noticing quite a few houses have bars in front of the windows.Q: Do you think that is to deter property crime?A: Oh, absolutely, absolutely....and possible bodily crime too, depending on the inhabitant, but the main reason would be for property crime, for this time of the day, when people are not at home.Q: What are indicators that makes you say it′s a poorer area?A: The way the houses are. There′s a lot of trash in the front and old furniture.A: If there were more people around, I would probably feel safer.Q: How do you like this neighborhood? Is it different?A: A little bit. It just seems like people are more aware of their surroundings and keeping their lawns and cars and everything together and nice.Q: Do you use public transportation?A: No. (the most common answer)Q: I see that everybody has cars at these houses, even though they do not look so nice.A: Another thing I′ve noticed is people parking in their yard, even when there′s a driveway.A: I think the assumption is in “nicer areas” you′re at less of a risk of crime.Q: Yes, that is the assumption.A: But unless there′s a neighborhood watch, we could easily park in front of a house, break into in, and be out in 5 min, if not less. That why I say there′s an “illusion of safety” for people here.

### 3.3. Results of Moments of Stress

All 46 test participants were equipped with Empatica E4 wristbands to collect physiological measurements during the SVG. These measurements were then fed into a newly developed algorithm to detect test participants’ moments of stress (MOS) [32], followed by a custom spatial analysis [19]. The main purpose was to identify locations along the route that would show participants’ MOS and whether these locations could be associated with contextual information from the environment and/or topics discussed in the geonarrative. The following two interesting observations were made, when MOS locations from the subset of ten participants (five female and five male) were overlaid over the traveled route (Figure 12). First, the main intersections along the route show MOS hotspots. Several strong hotspots are located at Florida Blvd/North 22nd St., North Boulevard/North 22nd St., Government St./Camelia Ave. One explanation could be that the observed high densities are due to reduced movement speed. Another hypothetical explanation may have been that participants, sitting next to the driver in the front of the vehicle, did not always trust the driver’s ability to drive safely and that this “mistrust” was heightened at major intersections. This finding should certainly warrant more in-depth analysis and attention in future research. It is also important to note that this algorithm for detecting MOS was tested for walking and biking, and not for driving in cars [32]. Another remark includes strong MOS hotspots on North 28th street, particularly next to a very large lot under construction (i.e., new school), while in the same area there is very high positive polarity from sentiment analysis. The beginning of the route shows reduced stress and strong cold spots were identified on the first two streets of the route.

The second interesting observation was made, when aggregated KDEs of MOS occurrences from all five female participants were separately mapped from all five male participants (Figure 12). While male participants articulated in the geonarrative that they felt fairly safe, especially when traveling through high urban blight neighborhoods, female participants seemed to be more uncomfortable with the same situation. As a general observation, the spatial distribution of male MOS shows more non-significant hot spots than female ones. Among the hot spot differences, we noticed one at the entrance of Capitol Senior High Highschool on North 23rd St., where one of the male participants was describing “Honestly, I think part of the reason some people get nervous around this area… if you compare other areas of the city, they have a much better lighting scheme”. Another difference is at the intersection of Saint Rose Avenue/Government street, where participants mentioned “[…] Here I was offered crack several times…” or “There is a change of house design right here to here”. Significant MOS hot spots for females were more predominant, while cold spots tend to occur in similar locations as the male ones. Two strong hot spot differences are emphasized: one along N 28th St. (which influenced the MOS detection for all participants, see Figure 12), and the second one along North 22nd St. For the first case, participants mentioned “It doesn’t look very nice, it is pretty plain […]” and, while answering the question regarding if blocked windows and other indicators of blight from this area attract crime “I think it does, because if you were in a more upkept neighborhood it is not as likely that there is a lot of crime”. Similarly, in the second case, one of the participants mentioned “It’s not upkept at all”, referring to the cemetery close by. These mixed results show high variability while traveling through all types of neighborhoods (i.e., high or low crime rates), without significant influence on a high occurrence of urban blight.

Video streams associated with MOS clusters showed the survey vehicle approaching a “Stop” sign of an intersection, oncoming vehicle traffic, or an approaching bicyclist. The survey vehicle seemed to also serve as a safe “haven”, when traveling through high urban blight neighborhoods since most participants responded that they would not feel safe walking in the same neighborhoods along the route, especially when it would be dark. Walking instead of driving inside a vehicle would have most likely increased the MOS density of test participants, especially in high urban blight neighborhoods.

## 4. Conclusions, Limitations, and Future Work

Mixed-methods approaches are increasingly adopted in Geographic Information Science and geospatially related work. The research presented in this article builds upon this development and applies different geospatial technologies to collect fine-scale information on physical urban blight occurrences and about peoples’ perception of safety issues. This work presents results on a reduced number of participants as proof of concept for the mixed methods approach. Combining quantitative and qualitative data from a mix of technologies is challenging and not without limitations [37,38]. Technologies include a standard questionnaire, spatial video, geonarrative, and biosensing wristbands. Data collected capture different ways people express their ideas and opinions (questionnaire and geonarrative), how such ideas and opinions are triggered/contextualized/influenced by what they see (spatial video), and simultaneously measuring peoples’ subconscious feelings (skin conductive wristbands). Responses are matched temporally and spatially and then they serve as input and are analyzed in a GIS using kernel density estimation and sentiment analysis.

There are two types of spatial and temporal lags that are worth discussing. The first one is dependent on the technology used and the combination with other tools. For example, Empatica E4 has a sample rate of 4 Hz (four samples per second), thus, when compared or merged with a device with different sampling relatively small lags can occur. Ref. [39] shows that “the local trend of the E4 sensor is lagging 2 and 1 s, respectively, “behind” the local trend of the VP sensor on average.”, where Empatica E4 was the wristband used in the present research and VP is another type of tool for measuring physiological parameters. Hence, there can be lags between instruments, as in our case between the video and the wristband measurements. Another type of lag, specifically for this research, can be related to the physical environment (cognitive lag 40). For some participants, it took longer than others to react to their surroundings because they were engaged in other storytelling. This will not be a strong lag if the field survey would have defined questions, instead of the semistructured version used for this manuscript. However, in this case, the qualitative side of the project would be reduced and loss of information would occur [40].

Subject selection can imply a bias in the research results [41]. Although the selection was purposeful, including participants who can enhance the understanding of perceived safety in this case study, the final list is not always balanced. In this work, we included 46 geonarrative participants, from which we selected 10 as proof of concept. The conclusions drawn from the selected participants may not be the same with a larger and more diverse population. The 10 selected individuals are students from two departments, and they could be biased due to various reasons, such as nervous behavior, underlying disease, weight issues, gender, social constraints, mental health conditions, and many others. Additionally, we acknowledge that the age range of students and their familiarity with the surrounding environment is not representative of an entire population. Another source of bias is based on the use of a predetermined route. Although multiple criteria were covered before defining the route, some elements may have been missed or not well balanced (such as high blight areas vs. low blight areas). Studies show that familiarity with an area is important in defining what is perceived as a safe place [42], thus a limitation in the present study involves the fact that some participants were more familiar with the area than others.

Results from the questionnaire indicated that most test participants felt less safe at gas stations, in alleys, and under bridges or underpasses. Data from spatial videos, geonarratives, and skin conductive wristbands were simultaneously collected from test participants sitting in a vehicle while driving with them along a pre-selected route. The route traversed through different residential neighborhoods in Baton Rouge, USA, with diverse socio-economic and ethnic populations. The results from the spatial video found a high density of physical urban blight occurrences along the northern and middle portion of the route and low urban blight density along the southern portion of the route (i.e., Garden District). In contrast to the diverse spatial distribution of urban blight density, reported crime densities were more equally distributed along the route.

Results from geonarratives and physiological measurements were based on a subset of ten participants (five women and five men) from a total of 46 test participants. Results from sentiment analysis conducted from geonarratives showed some unexpected results, with positive polarities indicating higher densities than negative polarities and both positive and negative polarities being found in the high urban blight density neighborhoods along the route. It was expected that most polarities in these neighborhoods would be negative. In contrast, and expectedly, exclusively positive polarity was found in the low blight density neighborhood along the southern portion of the route (i.e., Garden District). There were also many mixed sentiments extracted from all geonarratives and it is planned that future research will focus on creating categories or applying topic modeling to analyze such sentiments.

Physiological measurements were collected with Empatica E4 wristbands during the SVG, with some of these measurements being input into a newly developed algorithm to detect test participants’ moments of stress (MOS) [15,18]. The analysis showed that MOS hotspots occurred at major intersections along the route or approximately in the middle of long roads, concluding that test participants’ level of stress was impacted more by certain traffic patterns, rather than by perceived safety or high urban blight density occurrences. However, it should be noted that participants felt safe traveling along the route through high crime and high urban blight neighborhoods inside a vehicle, but would have felt unsafe walking along the same route, especially at dark. Walking (at dark) would have certainly increased the intensity of MOS of test participants, together with the darkness–light effect [43]. On the other hand, traveling inside a vehicle did not protect test participants from being involved in an accident, especially at major intersections. Another interesting observation was that while male participants, in contrast to female participants, expressed that they felt fairly safe traveling through high crime and high urban blight neighborhoods, surprisingly MOS analysis showed hotspots for both male and female participants. It will be interesting to find out, whether this apparent contradiction among male and female participants still holds, even after geonarratives and physiological measurement results of all 46 test participants have been analyzed.

This interdisciplinary research based on mixed-method approaches can be considered a step towards CPTED. Whilst at the beginning the approach was considered all about the building environment, nowadays it is important to introduce the social environment. When designing a CPTED methodology researchers need to ensure dealing with the safety perception parameter. Through this work, we identified elements from both categories, such as spatially determining urban blight locations and analyzing participants’ perceptions while traveling through a mixed environment. As an application to CPTED, the use of spatial video and geonarratives (SVG) can be introduced for understanding communities, how their environment plays a role in their neighborhood, to what degree social cohesion is important for their safety perception, while the moments of stress from the physiological measurements complement the safety or unsafe feelings of the participants expressed through their voice. In this way, targeted urban design interventions can be introduced by focusing on changing environmental factors. CPTED aims to include the community to make effective decisions regarding urban planning strategies to optimize the quality of life within specific neighborhoods.

Researchers have been discussing the law enforcement members’ perceptions and neighborhood citizens’ perceptions of crime through two different lenses without comparison or trials of determining community policies [44]. Whilst only presenting results of a sample of participants, the study design included students, local stakeholders, experts in blight, and criminal activities issues. Additional participants are needed to embed an equal number of members for each group and compare relationships in future work.

## Figures and Tables

**Figure 1 ijerph-18-01284-f001:**
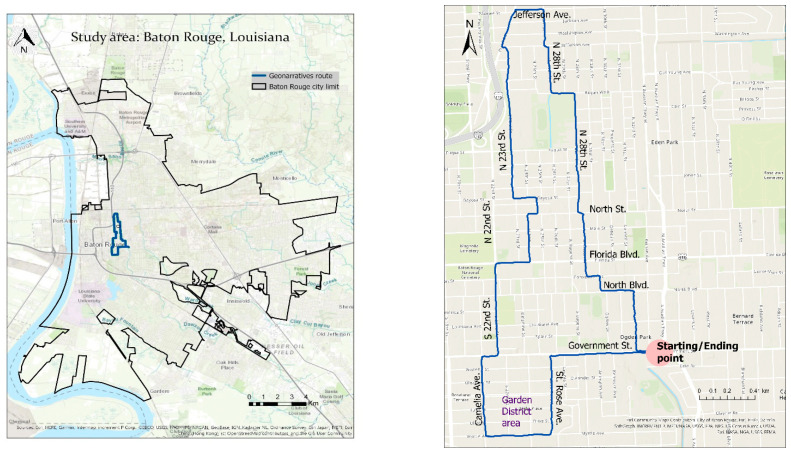
The city of Baton Rouge and the selected route (outlined in blue) that was driven by researchers and test participants for fine-scale geospatial data collection (**left side**) and the detailed route (**right side**).

**Figure 2 ijerph-18-01284-f002:**
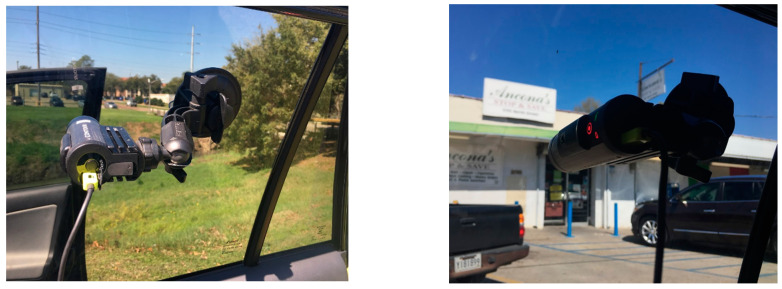
Examples of spatial video cameras mounted to the inside window of a vehicle.

**Figure 3 ijerph-18-01284-f003:**
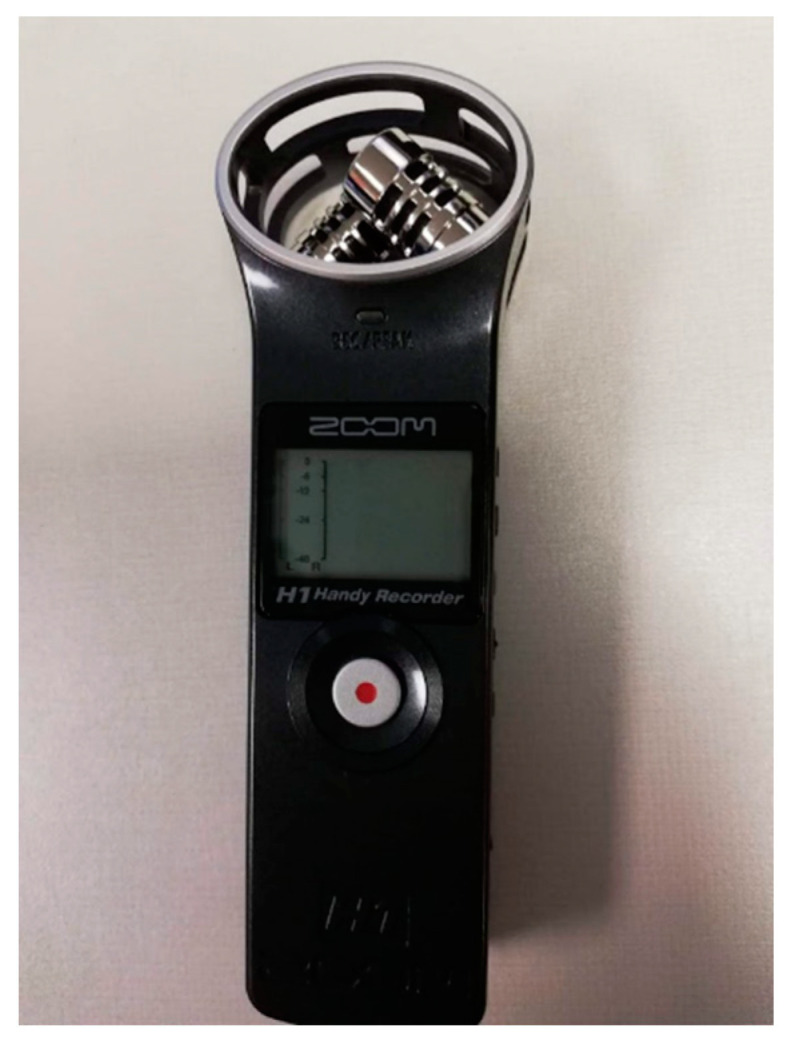
Audio recorder used to collect geonarratives.

**Figure 4 ijerph-18-01284-f004:**
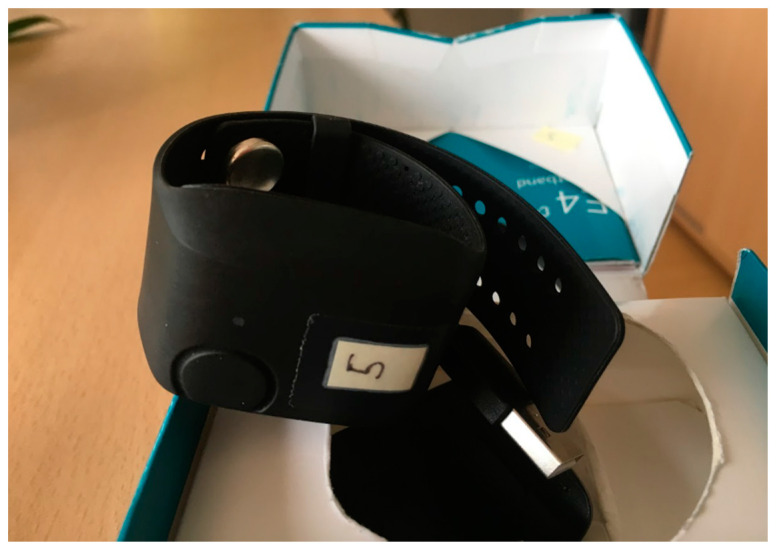
Wearable sensor collecting psychophysiological parameters during spatial video geonarrative (SVG).

**Figure 5 ijerph-18-01284-f005:**
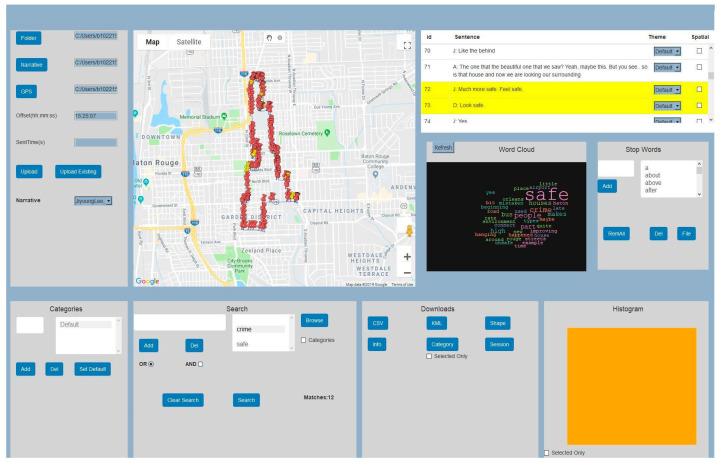
The interface of the software Wordmapper: Reading the input data (**upper left**); showing the mapped route with pins (**upper middle**); displaying individual sentences, a word cloud, and stop words (**upper right**); additional filters, queries, and histogram display (**bottom**).

**Figure 6 ijerph-18-01284-f006:**
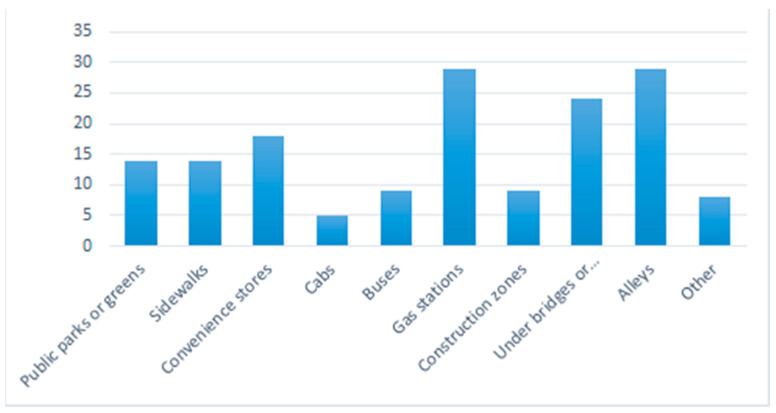
Places where test participants felt less safe in Baton Rouge, USA.

**Figure 7 ijerph-18-01284-f007:**
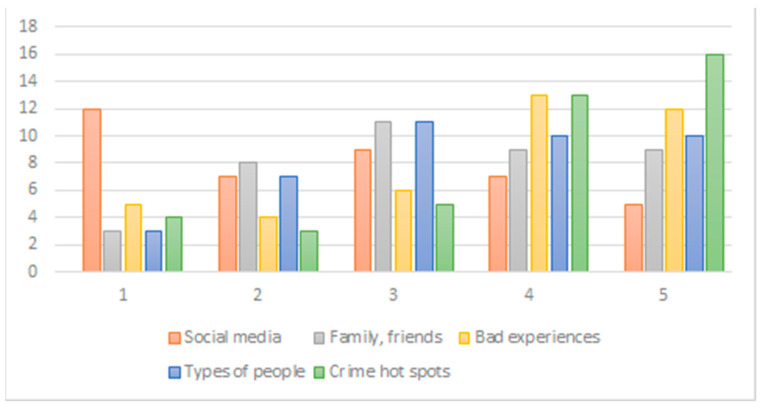
Categories influencing test participants’ crime perception from 1 (not at all) to 5 (most).

**Figure 8 ijerph-18-01284-f008:**
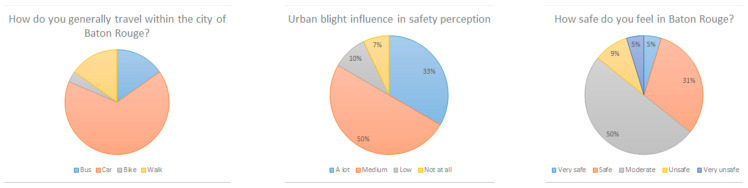
Participants’ general mode of travel (**left**); the influence of urban blight on safety perception (**middle**), and feeling of safety in Baton Rouge, USA (**right**).

**Figure 9 ijerph-18-01284-f009:**
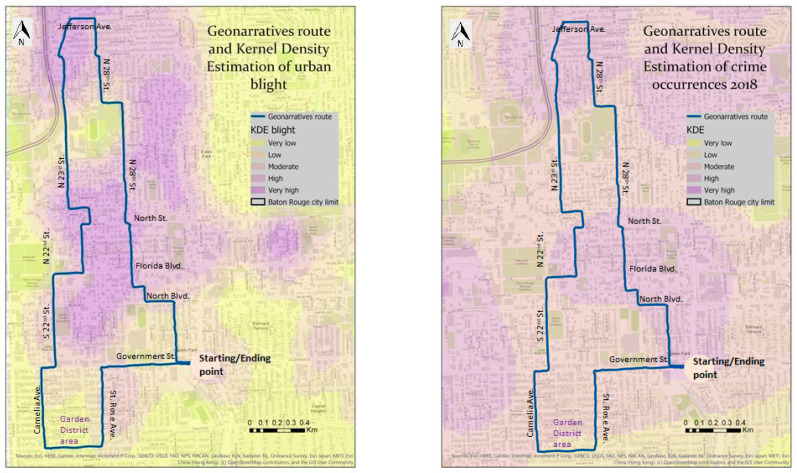
Route traveled by test participants overlaid on top of kernel density estimations (KDEs) of physical urban blight occurrences (**left**) and of reported crime locations (**right**).

**Figure 10 ijerph-18-01284-f010:**
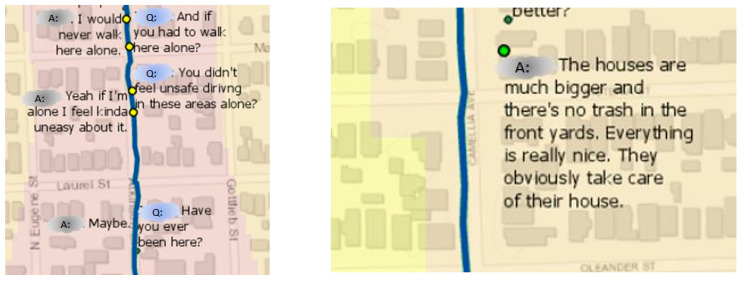
Example of mapping a portion of a geonarrative’s transcript applying Wordmapper (Ajayakumar, Curtis et al. 2019).

**Figure 11 ijerph-18-01284-f011:**
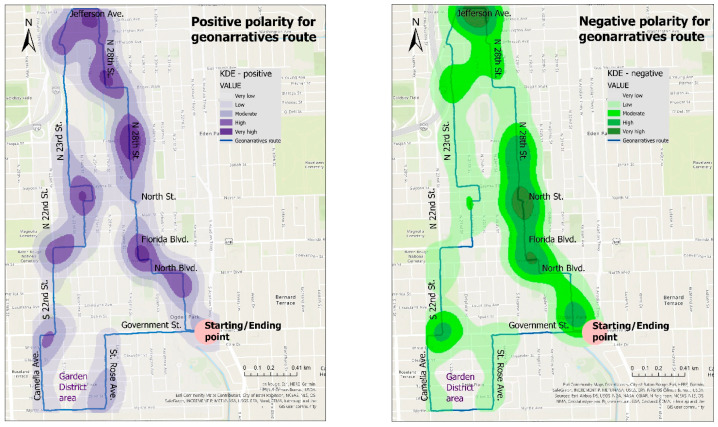
Route traveled by test participants overlaid on top of results of sentiment analysis based on transcripts of ten geonarratives: KDEs of positive polarity (**left**) and negative polarity (**right**).

**Figure 12 ijerph-18-01284-f012:**
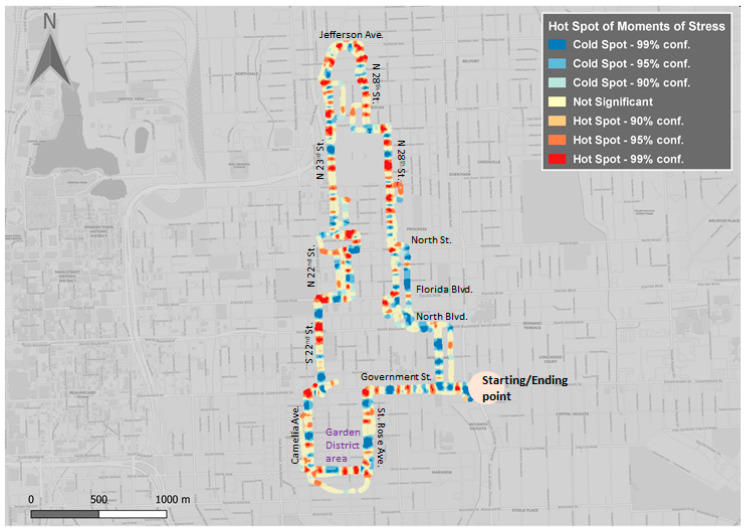
Spatial distribution of moments of stress (MOS) of all participants (top), five Figure 4.

## Data Availability

The data presented in this study are available on request from the corresponding author. The data are not publicly available due to the large storage volume of all spatial video geonarratives collected in this study (approx. 1TB) and due to privacy concerns of physiological measurements collected from individual participants.

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
