# Peer review of "Applying Spatial Video Geonarratives and Physiological Measurements to Explore Perceived Safety in Baton Rouge, Louisiana"

_ijerph, 2021, doi:10.3390/ijerph18031284_

Round 1
Reviewer 1 Report
Major problems
1. The authors use different methods (e.g., standard questionnaire, spatial video, geonarrative, and biosensing wristbands) to obtain information, then classifies regions as "safe" or "unsafe", however these methods consist of different spatial resolutions, thus are not easy to conduct comparison. Moreover, these attributes only represent perceived safety. How can a fair comparison be conducted among each of these methods? Also, how can these results be associated with actual crime figures within the city?
2. Line 122: What do you mean by "in the control of researchers"? If that is the case, the current research is subjective in nature, thus may not have broad applications into other areas (especially those with different transportation framework and urban planning structure)
3. Line 126-128: Provide some explanations of the applications - in 1-2 statements for each.
4. Line 159: What do you mean by "very environment", do you actually mean "adverse environmental conditions"? If so, better provide some descriptions
5. Section 2.4.2: The purpose of using SVG is still not clearly mentioned and described. In particular, what information and attributes will it provide? Is it subjective enough for long-term and extended spatial assessment?
6. Line 176-177: describe such algorithm briefly, how is it operated, and what tools do it use for data analysis?
7. Line 289-301: How about intermediate statements? How to classify it ? It's better to formulate a scoring table (say from 1-10, with 10 representing the most positive and 1 representing the most negative statement), and allocate a score to each of the collected statement. The authors have assumed statements can be either positive or negative, i.e., either black or white - however there are grey, yellow, green...sides.
8. Line 306: Are 10 transcripts sufficient?
9. Better label all streets (mentioned in the article) in Figures 9, 10, 11 and 12 respectively. Otherwise, it will be unfamiliar and inconvenient for readers outside US. e.g., Government St., Garden District, Florida Blvd/North 22nd St, North Boulevard/North 22nd St, Government St/Camelia Ave
10. Why the result is as expected in Line 313? Any environmental or spatial reasons or factors?
11. Line 324-325: How do it depend on car speed, as well as environmental conditions? Better provide some explanations regarding the inter-dependence of all factors.
12. Figure 12: Results are obtained solely on 46 test participants - are the results representative? Also, how reliable are the spatial plots here? The experiment involves personal opinion, and no validation is done, therefore current results may not be too promising nor trustworthy.
Minor problems
Some typos / grammatical mistakes made:
Line 56: Better use (1), (2) and (3) instead of "first, second etc."
Line 68: This research is based on handling of data acquired...
Line 85: asked --> required
Line 87: mapping exercise were not
Line 99: by applying spatial video technologies
Line 107: the driving time should last for around 30 minutes
Line 116: videos were
Line 118: demand--> require
Line 119: video camcorder, which allows the collection
Line 125: At present --> Currently
Line 136: pointing towards the driving direction
Line 139-140: eliminated noise effects inside the vehicle environment
Line 170: psychological attributes
Line 253: 6 out of the 44 respondents
Line 257: Around 2/3
Line 282: The current boundary
Line 289: were applied to analyze such text
Line 312-313: lacked
Moreover, the authors should use commas in a more coherent and consistent manner, for example:
Line 69: mapping exercise, and a field survey...
Line 83-84: were set up online, and participants were expected to complete the forms and submit their responses to the authors of this research.
Line 86-87: unsafe, and to provide a short description of the perceived feelings.
Line 88-89: each lasted for about 10 minutes, and details can be found in [8].
Line 95: , and selected based on...
Line 100-101: locations, with each location
Line 109: Figure 1, and it is located almost in the middle of Baton Rouge
Line 161-162: likely impossible, when the interview is conducted in an office space
Reviewer 2 Report
Please see the attached document. Thank you!

Reviewer 3 Report
The authors present a very interesting extension of the SVG methodology. The addition of physiological measures to geo-narratives provides a fascinating opportunity to capture unstated responses.
A couple areas where the paper may be improved:
The background section of the manuscript could be improved by a more expansive coverage of the CPTED and fear-of-crime literature. The manuscript is indeed a novel contribution to both and should be placed within the body of literature more definitively.
The dynamic interplay between subjects and their environment seems to focus solely on the physical environment. It's not clear what other factors were collected that may affect perceptions of safetly. For example, were street corners with groups of young men who may have been perceived as dangerous coded? The issue of racial differences between subjects and those in the viewed community needs to be addressed. (Age differences as well.) The literature on fear of crime should inform this section.
Overall, this is a very novel approach to the use of SVG that addresses an interesting topic.
Reviewer 4 Report
Review. ijerph-1038099. Applying Spatial Video Geonarratives and Physiological Measurements to Explore Perceived Safety in Baton Rouge, Louisiana. The article investigated the integration of Spatial Video Geonarrative and physiological measurements, and its application in linking SVG’s semantic values and individuals’ stress situations. The research objectives are important and clear, the state-of-the-art methods are used, the data collected for analysis is appropriate, and the conclusions are reliable. The article is appropriate for the journal IJERPH. I suggest a minor revision before acceptance. I list my comments below:
- As the authors indicated that, the selection of the route was crucial for this research. I wonder the representativeness of the selected route in this study. If future study wishes to expand the research area, does the researchers need to drive through more streets? Is there any way to make sure future study have similar number of samples, even its study area is way larger than Baton Rouge?
- Readers may not have enough background knowledge to understand the link between GSR, ST, BVP and individual’s motion. Some theoretical hints and practical literature would be helpful.
- I wander how to avoid the impacts of inherent physical conditions of 44 participants on results. By this I mean some participants may be easy to be nervous or frightened. Is there any procedure to control this issue?
- When visualizing reported crime and blight cases, regular KDE was used. Considering the selected route in this study, may be KDE along network is more suitable?
Reviewer 5 Report
This is an interesting and exciting work. Researchers applied new technologies (i.e., video and sensors) to the environment safety measurement. It provides a new vision in many research fields. Here, I list several suggestions, which may help improve this research.
- Please have a comprehensive literature review. Are there any related studies? What are the pros/cons of other studies? For example, so far as I know, many researchers are investigating the relationship between crimes and the built environment (i.e. ). What are the cons of those studies, and what will be the contribution of this research?
[1] Wilcox, P., Quisenberry, N., & Jones, S. (2003). The built environment and community crime risk interpretation. Journal of Research in crime and delinquency, 40(3), 322-345.
[2] Kitchen, T., & Schneider, R. H. (2007). Crime prevention and the built environment. Routledge.
[3] Yue, H., Zhu, X., Ye, X., & Kudva, S. (2018). Modelling the effects of street permeability on burglary in Wuhan, China. Applied geography, 98, 177-183.
- In section 2.1, I suggest describing more about crimes, such as crime types. If there are many data applied in the study, list them in a table would be better.
- Could you also please clarify more about participants if you have the information, i.e., occupation, age, how long they have stayed in the study area? Besides gender differences, these factors impact their perceptions of safety as well.
- Could you give a demonstration of the questionnaire and answers from participants? There are only a few samples in figure 10.
- From my point of view, crimes are closely related to the environment. However, the different environment may attract different types of crimes. So, it would be better if authors can differentiate crime categories when analyzing the results.
- In the results, I didn’t see the correlation between safety and video content. If authors can extract objects (green space, road, sky…) and explore relations to the survey results in future work, it will be very interesting.
- I have a suggestion which may improve your future research. Streetview images are great sources for understanding urban environments. There have been many studies working on it. Authors can get those images from Google API, which may help you simplify the data collection process. In the meantime, authors can have a comparative study based on geo-video and streetview images. One source is dynamic, and the other is static, so there are might be some differences when participants give their feedback about safety perception.
Author Response
Please see the attachement.

Round 2
Reviewer 1 Report
Thank you for writing a detailed reply based on our questions. It's very nice and meaningful.
Referring to our previous questions and your respective replies, we have some further comments or suggestions for improving current manuscript:
Q1: How can audio data archived be re-analyzed in GIS-related environment? According to my understanding, GIS is only capable of accepting particular format of datasets, and has restriction on file format, raster/vector etc. Could you explain further on this issue, as well as the practical applications of questionnaire and audio results?
Q1: Complementing information at a fine-scale on perceived safety is conducted, and the methodology is acceptable, however, how can you use it for comparing with real crime data analytics? It's sound not straight-forward.
Q5: It would be much better to give some details of spatial video methods within the manuscript - at least the use of SVG is a bit unfamiliar for common readers.
Q7: If the system is only capable of distinguishing positive statements and negative statements, then its effectiveness and applicability is quite low - because most of the uttered statements and replies in daily lives are either neutral, or without sentiment, or even do not consist of personal feelings. How do you assess the effectiveness of the system for practical daily usage? Any proper evidence that shows better linkage between the "utterance" of citizens with potential crime incidents - some people may say things in a positive manner, however they are extremely worried, and the accuracy of current system has to be further assessed - based on advanced technology, which can be used to "analyze" the actual meaning of a statement / utterance. Solely relying on the key phrases of an utterance is basically wrong for some cases.
Q12: Thanks for pointing out 3 previous literature on similar studies. It is well-understood that finding sufficient participants for experiment / study is rather hard, however, conclusions may not be the same if there are more participants involved in the same study, therefore we suggest the authors check the entire manuscript once again, and replace strong words like "must", "have to be", "is/are" by "may", "could possibly" and "should". It will be much better. On top, some of the potential errors / gaps of current study should be explicitly written / pointed out in Section 4, as limitations of current studies.
Overall, it would be better if the participants involved in this study are from different age ranges. On top, more participants will help to confirm the validity of some of the tests adopted in this study. Will that be possible to include results of more participants within this manuscript? If so, it will be fantastic.
Reviewer 2 Report
Thank you for your thoughtful revisions and attention to detail. At this point, I have only a few minor points that should be addressed:
1) Abstract: The revised version provides much more information to the prospective reader. Thank you. However, in lines 28-31 there is more detail than necessary and, in my opinion, detracts from the summary nature of the abstract. Please remove the following text: “however, as similar, the topics discussed in the northern part of the route include signs of urban blight. There is no immediate explanation from the videos for the detected MOS although many of them are detected when the car approaches the signs for the intersection or when a cyclist is approaching.”
2) Line 61: What is the meaning of the following term: “physical deterministic design”. Please choose a term or description that will be more readily understood by the wide readership of this journal. Perhaps “built environment” would be sufficient?
3) Line 236: Do you mean Fig. 1 in the following text? “A SVG was completed with all test subjects while travelling along the same route as depicted in Figure 3.” However, Figure 3 is “An audio recorded used to collect geonarratives”.
4) Figure 10: The map on the left has the following text which is not useful and should be removed:
“Q: Have you ever been here? A: Maybe”
5) Figure 11: It is difficult to visually investigate the patterns in both maps as the KDE output is in the same color. Please revise to improve visual comparison, for example, present the negative polarity as red and the positive polarity as green.
6) Lines 487- 489: Thank you for addressing limitations in this study. In these lines you list a number of variables that can introduce bias into your participant sample. Please make sure you include the appropriate references that support this text.
